# Genetic Variation and Preliminary Indications of Divergent Niche Adaptation in Cryptic Clade II of *Escherichia*

**DOI:** 10.3390/microorganisms8111713

**Published:** 2020-10-31

**Authors:** Zhi Yong Shen, Xiu Pei Koh, Yan Ping Yu, Stanley C. K. Lau

**Affiliations:** 1Department of Ocean Science, The Hong Kong University of Science and Technology, Clear Water Bay, Kowloon, Hong Kong, China; zshenae@connect.ust.hk (Z.Y.S.); yyuaz@connect.ust.hk (Y.P.Y.); 2Division of Environment and Sustainability, The Hong Kong University of Science and Technology, Clear Water Bay, Kowloon, Hong Kong, China; xpkoh@connect.ust.hk

**Keywords:** *Escherichia*, cryptic clades, ecotype, bitscore, genetic variation

## Abstract

The evolution, habitat, and lifestyle of the cryptic clade II of *Escherichia*, which were first recovered at low frequency from non-human hosts and later from external environments, were poorly understood. Here, the genomes of selected strains were analyzed for preliminary indications of ecological differentiation within their population. We adopted the delta bitscore metrics to detect functional divergence of their orthologous genes and trained a random forest classifier to differentiate the genomes according to habitats (gastrointestinal vs external environment). Model was built with inclusion of other *Escherichia* genomes previously demonstrated to have exhibited genomic traits of adaptation to one of the habitats. Overall, gene degradation was more prominent in the gastrointestinal strains. The trained model correctly classified the genomes, identifying a set of predictor genes that were informative of habitat association. Functional divergence in many of these genes were reflective of ecological divergence. Accuracy of the trained model was confirmed by its correct prediction of the habitats of an independent set of strains with known habitat association. In summary, the cryptic clade II of Escherichia displayed genomic signatures that are consistent with divergent adaptation to gastrointestinal and external environments.

## 1. Introduction

Closely related bacterial lineages can have very different habitats and niches; ecological differentiation was reported between Vibrionaceae strains coexisting in coastal ocean [1], as well as between typical *E. coli* (host-associated) and environmental cryptic clades of *Escherichia* [2,3]. Furthermore, multiple ecotypes often exist within traditionally defined species. In fact, the survival and distribution of a species were partly influenced by intra-species diversity [4]. For example, Bacillus simplex ecotypes were adapted to different microhabitats in the “Evolution Canyons” in Israel [5], while different host range was observed among *Legionella pneumophila* [6] and among *Salmonella enterica* [7,8]. Identifying and characterizing bacterial populations with distinct ecological niches (ecotypes) has been fundamental to understand their ecology and evolution [9,10].

Many studies delineate ecologically distinct populations based on macroscopic characteristics such as gene absence/presence and pseudogene analysis, others focused on finer scale differentiation, examine impacts of gene mutations and indels on protein function [8]. The bloom of “omics” analyses—genomics, transcriptomics, proteomics, phenomics, etc.—with many valuable insights particularly attained from the ever-expanding collection of bacterial genome sequence data substantially benefited our understanding of ecological differentiation and niche adaptation of bacteria [3,8,11,12,13]. Attempts to gain further functional insight into the underlying biological mechanisms of niche adaptation among closely related bacteria has been greatly facilitated by artificial intelligence approaches such as machine learning, which added invaluable depth and possibility to the interpretation of massive and complex genomic data. Combining data across whole genome or proteome, prediction of phenotypes from genotypes, and identification of genetic signatures of niche adaptation in pathogenic bacteria were made possible by machine learning, which otherwise would be difficult if not impossible to execute with other methods [13,14,15,16]. 

Typically, the identification and characterization bacterial populations starts with identifying distinct genotypic clusters [9]. The distinctness of putative ecotypes could then be validated by the distribution of microhabitats, difference in genome content, gene expression profile, physiology, and phenotypes, as ecotype adaptation to habitat and ecological niche are often reflected in these aspects [1,3,8,10,11,12]. Source and virulence attributes of isolates could be predicted based on their inferred phenotypes and the association of phenotypes with ecological attributes such as host type or range, which is especially invaluable for public health management [13,16].

In lineages that have diverged recently, disruptive frameshifts, truncations and complete deletion of genes have yet to occur. Instead, mutational (or small indels-induced) function loss or alteration occurred as an immediate response to a new environment [17]. These changes would be overlooked by macroscopic approaches but could be detected by a recent.delta-bitscore (DBS) approach that identify functional divergence induced by genetic variations could detect them [8]. DBS successfully identified signatures of host adaptation among different *Salmonella enterica* serovars, showing consistency with pseudogene analyses but with additional sensitivity in detecting protein variants that are functionally altered or non-functional [7,8]. It is important to discover the types of functional divergence favoring rapid adaptations that could support the coexistence of most closely related yet ecologically distinct populations (i.e., within a named species). Probing such fundamentals of adaptation for their coexistence would ultimately advance our understanding of the functions of different ecotypes, their complex interactions within bacterial community, and with their habitats [10].

Our group has published the genome sequences of 16 strains affiliated with cryptic clade II of *Escherichia* [18]. The strains were isolated from intertidal sediment, whereas prior to the publication strains belonging to this lineage were rarely isolated and only a handful of genome sequences, all of fecal origins, were publicly available [19,20,21,22,23]. Therefore, many of their ecological attributes were yet to be uncovered. As clade II strains were found in contrasting habitats, and strains from fecal sources and coastal marine sediment were phylogenetically more distantly related [18], we proceed to examine if clade II is comprised of strains that were ecologically divergent. Capitalizing on the capability of computational approaches to infer metabolic capacities from genomic data, we subjected the genomes of selected strains to DBS analysis [8] to detect functionally significant genetic variations. Utilizing bitscore difference as the input, we adapted the random forest model from Wheeler et al. [13] to detect signals of niche divergence in selected *Escherichia* cryptic clade II strains, identifying genes that displayed signs of functional divergence, which corresponded to their niche adaptation to hosts and external environments, respectively. The findings could facilitate the selection of candidate genes and functions for further ecological studies to confirm their ecological distinctness.

## 2. Materials and Methods

### 2.1. Strain Selection and Genome Data

Six *Escherichia* clade II strains were compared in this study. Three clade II strains (E4742, E4385, and E4930) isolated from coastal marine sediment in our previous study [18] were selected for PacBio sequencing. Another three clade II strains (B1147, ROAR019, and EC5350) were of fecal origins. [20,23,24], and their genomes were retrieved from the NCBI database. Genomes of other *Escherichia*, including *E. coli* (ECOR66), cryptic clade I (E. TW15838 and E. TW10509), clade III (E.TW09276 and E. TW09231), clade IV (E. TW14182 and E. TW11588), clade V (E1118 and E. TW09308), and *Shigella dysentriae* (Sd197), were also retrieved for machine learning training and assessment. Accession numbers and information of the genomes can be found in Table A1 in Appendix B.

### 2.2. Genomic DNA Extraction

Genomic DNAs of the three clade II strains (E4742, E4385 and E4930) were extracted from overnight pure cultures cultivated in LB broth using Qiagen RNA/DNA Mini Kit (Qiagen, Hilden, Germany).

### 2.3. PacBio Sequencing

For each genomic DNA sample, a 10 kb SMRTbell library was constructed according to manufacturer’s instructions (PacBio, Menlo Park, CA, USA). Each sample was first sheared and treated with Exo VII, followed by the DNA damage repair and end-repair steps. Following end-repair, barcoded adapters are ligated to the sample. Samples were then pooled and underwent Exo III and VII treatments, followed by two 0.45X AMPure PB bead purification rounds. The libraries were sequenced using a PacBio RSII instrument and P6C4 chemistry (PacBio, CA, USA).

### 2.4. Genome Assembly Assessment and Annotation

CANU assembler V1.5 [25] with the standard pipeline and default parameters was utilized to create assembly from PacBio long-reads, resulting in two draft genomes (E4835 and E4930) and one complete genome (E4742). The genomes were submitted to Dfast web server (https://dfast.nig.ac.jp/) for annotation. Generated gff files and predicted protein files were used for subsequent comparison study. The completeness of genome assembly and annotation for all genomes were assessed with the Benchmarking Universal Single-Copy Orthologs (BUSCO) V3.0 software [26]. OrthoDB v9.1 “enterobacteriales_odb9” base was used as a reference (781 BUSCOs among 216 species of the order Enterobacteriales). Each individual genome was examined for the copy number of 781 BUSCO genes (single-copy orthologues present in at least 90% of enterobacteriales). For each BUSCO gene, a consensus protein sequence was produced using HMMER [27] hidden Markov models (HMMs) and then used as search query against each genome to identify up to three putative genomic regions by tBLASTn [28,29]. The de novo gene structure of putative genomic regions was then predicted by AUGUSTUS [30]. These predicted genes were aligned to HMM alignment profile of the BUSCO gene and only those with alignment bitscore higher than cutoff value (90% of the lowest bit-score among reference genomes) were kept. If no predicted gene from a specific genome was retained, the absent gene is assigned as “missing.” Genes with aligned lengths shorter than 95% of the expected BUSCO group lengths were classified as “fragmented.” Predicted genes were classified as “complete” if only one copy was present in a genome or “duplicated” if more than one “complete” predicted genes were present in a genome.

### 2.5. DBS Calculation and Hypothetically Attenuated Coding Sequences (HACs) Definition

We retrieved the Pfam HMMs collection from ftp://ftp.ebi.ac.uk/pub/databases/Pfam/current_release and use the curated Pfam-A.hmm database in this study. Annotated protein sequences (from Section 2.4) were aligned to corresponding profile HMM in the Pfam database using hmmsearch from the HMMER3.0 package (http://hmmer.org) to produce bitscore values for each query sequence.

When comparing the proteomes of two strains, for two orthologous sequences (each from one strain) aligned to the same profile HMM, delta-bitscore (DBS) was calculated following Equation [8]
DBS = Xref − Xvar(1)
where Xref and Xvar each stand for bitscore of reference strain and variant strain, respectively, which come from the alignment to the same profile HMM in Pfam database using hmmsearch from the HMMER 3.0 package.

Subsequently, for pairwise comparison of two proteomes, loss-of-function mutations were identified using empirical distributions of DBS for all orthologues, with 2.5% of DBS on the least dispersed end as cutoff as described in Wheeler et al. [8]. Proteins with scores falling outside the cutoff are considered as HACs, i.e., hypothetically attenuated coding sequences (HACs). For concurrent comparison of multiple proteomes, as in Wheeler et al. (2016), bitscores from all compared proteomes were first sorted. Then, the median score was used as a benchmark to compare the bitscore of each gene to obtain DBS. Subsequently, HACs were identified from DBS distribution using the same cutoff as the pairwise comparison above (2.5% of DBS on the least dispersed tail). The Wilcoxon signed rank test and one-way ANOVA were performed to test for significant differences in the degree of functional loss between two proteomes and between two groups of proteomes (gastrointestinal vs. environmental), respectively. Analyses were done using the Minitab R v18.1 software (Minitab Inc., State College, PA, USA). Boxplot for ANOVA was created via the ggpubr package in R.

### 2.6. Random Forest Classifier Constructing and Training

As the clade II strains were from two isolation sources with disparate environmental conditions (external environment vs gastrointestinal), we proceed to train a random forest classifier model to separate the strains based on their isolation sources, which presumably would return a set of genes that were indicative of adaptations to the different habitats. To minimize the inclusion of genes that were phylogenetic markers rather than informative of phenotype, we included another three more distantly related genomes, i.e., *Shigella dysenteriae* Sd197, *E. coli* ECOR66 (both gastrointestinal), and *Escherichia* clade IV TW14182 (environmental). Previous studies on these genomes showed evidences of adaptations to either external environment or gastrointestinal habitats for the genomes [3,21,22,31,32,33,34,35,36]. Functional importance of sequence variations were scored using the DBS metric by comparing the protein coding genes of each strain to the profile HMMs of Gammaproteobacterial proteins (gproNOG.hmm) from the eggNOG database (http://eggnogdb.embl.de/). As we have performed with Pfam A in above pairwise comparisons, each protein sequence was searched against the gproNOG.hmm database using hmmsearch from HMMER3.0 package.

In this study, orthologs were selected using filtering criteria similar to Wheeler et al. (2018) [13]. We trained the random forest model on a set of 6709 orthologous genes to differentiate strains of external environment and gastrointestinal origins, with the performance of model assessed by out-of-bag (OOB) accuracy. The random forest classifier was built and trained using the R packages “randomForest” and “caret.” Tree parameters were tested to evaluate the best combinations. Number of trees (ntree) were set at 10,000 to optimize mtry (number of genes randomly sampled at as candidates at each node) as OOB error rate stabilized at this ntree. We tested different values for mtry (1, n/10, n/5, n/3, n/2, and n, where n = the number of predictor genes) and decided on mtry = n/10, as it would reduce the chances of sampling correlated predictors and with lower OOB error (OOB error = 0.2).

Model performance was sequentially improved through several rounds of iterative feature selection [37]. At the first iteration (model 1), all predictor genes were included in building the model, followed by sparsity pruning of predictors with variable importance (VI) ≤0. The model building and pruning rounds were repeated (rebuilding model with pruned predictors set, followed by pruning of predictor genes with lowest 50% of VI) until perfect OOB accuracy was obtained. We performed permutation testing to test the null hypothesis of random association between the predictor genes and response variables (gastrointestinal or external environment). Model building pipeline was repeated on 1000 permutated datasets where the response variables were randomized. The *p* value, i.e., the frequency of models with the same accuracy as that of the original data was determined. The final model was used to predict the isolation source of another set of seven cryptic Escherichia clades genomes with known isolation sources to assess the accuracy of the trained classifier model. CIII (E. TW14182 and E. TW11588) and CIV (E.TW09276 and E. TW09231) strains showed signatures of adaptation to aquatic environments, CI (E. TW15838 and E. TW10509) strains were enteric, and CV (E1118 and E. TW09308) strains have been reported to retain the ability to survive in the external environments without loss the ability to in persist in gastrointestinal tracts [3,21,22,31,33,34,36]. The top predictor genes of the best model were assigned to functional categories based on COG and KEGG database annotation matching via online web server KOBAS3.0 (http://kobas.cbi.pku.edu.cn/anno_iden.php).

Summary of the pipelines used to obtain DBS and to construct the random forest classifier can be found at https://gist.github.com/szypanther/0cd83513e07aa9e1f9929f5df9214864.

## 3. Results

### 3.1. Genome Assembly, Annotation, and Completeness

We achieved about 800 MB of PacBio raw data for the three strains (E4385, E4742, and E4930) from coastal marine sediment. The total coverage ranged from 151× to 179× (with assumed genome size of 5 Mb) after assembly, we obtained two draft genomes (E4385 and E4930) and one complete genome (E4742). The summary of raw data and assembly statistics are shown in Table A2 and Table A3 respectively. We first annotated the genomes and then assessed the completeness of assembly in terms of gene contents using the BUSCO set of 781 universal single copy orthologs found among 216 species of the order Enterobacteriales. The genomes had at least 93.7% of the BUSCO genes as complete genes while not more than 6.2% of the genes were missing (Figure A1).

### 3.2. Pairwise Comparison of Protein Function Loss among Six Cryptic Clade II Strains

DBS corresponded to difference in bitscores of each pairwise comparison of orthologous proteins between two cryptic clade II (CII) strains. Distribution of DBS values for each pair of strains was then used to infer functional loss in orthologues, with 2.5% of DBS on the least dispersed end as cutoff as described in Wheeler et al. (2016) [8]. Proteins with scores falling outside the cutoff are considered as HACs, i.e., hypothetically attenuated coding sequences. A complete table of DBS values and HACs can be found in Appendix A. We tested if each proteome pair has similar rate of function loss (Wilcoxon signed rank test) and observed no statistically significant difference (*p* value > 0.05) among the gastrointestinal strains (B1147, ROAR019, and EC5350) and among the external environment strains (E4385, E4742, and E4930) (Table 1). However, pairing of strains from different sources generally showed significant discrepancies in the rate of functional degradation (*p* < 0.05, except for comparison between B1147 and E4385 where *p* = 0.05).

### 3.3. General Trend of Gene Degradation in Gastrointestinal Cryptic Clade II Strains

As pairwise comparisons of DBS indicated a possible trend of divergence among the gastrointestinal and environmental strains, we further determined which group had higher degree of function loss (i.e., higher percentage of orthologs being classified as HACs). Bitscores of 2844 orthologous genes from each strain were extracted and the median score of all strains was used as benchmark to compare the bitscore of each gene to obtain DBS and subsequently determine HACs (Appendix A). The result showed that gastrointestinal strains not only generally had lower bitscores than the environmental strains. Moreover, the gastrointestinal strains also had significantly larger number of HACs than environmental strains. One-way ANOVA test display the significant difference of HACs number between the two groups (*p* = 0.008) via ggpubr package in R for data visualization (Figure 1A,B).

### 3.4. Classification of Gastrointestinal and External Environment Strains Based on Informative Genes

We build a random forest classifier to differentiate strains according to their isolation source (external environment or gastrointestinal origin), returning a set of interpretable predictor genes that were indicative of adaption to each habitat. Bitscore values of orthologous genes were used as input for training the random forest model, with the performance of model assessed by OOB accuracy. Model performance was improved by iterative feature selection, where predictor genes with VI = 0 pruned after initial model training, followed by repeated rounds of model retraining using top 50% of predictor genes until perfect OOB accuracy (100% accuracy) was achieved (Figure 2A). In the first round of model building, the whole set of 6709 orthologous genes that fulfilled selection criteria were utilized for training, returning 200 genes with VI that were noticeably higher than remaining genes (Figure 2B). On the contrary, 3873 orthologous groups had zero VI (variable importance = 0, i.e., they did not improve the accuracy of model or were left out by the model) and were not used in the first bout of feature selection (i.e., model 2). The sixth model achieved a perfect classification accuracy for source prediction. Model 6 is thus chosen, with 164 top predictor genes that were most informative for distinguishing the two groups of strains (Appendix A). Heatmap showing the clusters of the 164 genes based on their bitscore matrix value. (Figure 2C).

We failed to reject the null hypothesis that the association between the predictor genes and the predicted outcomes (association with external environment or gastrointestinal tracts) are random using permutation test (*p* = 0.401). Nonetheless, we noticed that for models built with permutated dataset that achieved the same OOB accuracy as the model built with original dataset (sixth model iteration, 100% OOB accuracy), proportions of majority votes were lower than that of the original model (see Table 2 for example).

To further confirm the accuracy of the classifier model, we applied the model on a collection of seven reference cryptic clade strains (with known isolation sources) that were unseen by the model before (Figure 3). In agreement with other studies, the classifier correctly predicted the sources of these strains [3,21,22,31,32,33,36]. The CIII (E.TW14182 and E.TW11588,) and CIV (E.TW09276 and E.TW09231) strains have been identified as external environmental strains. The CI strains (E.TW15838 and E.TW10509) had also been identified as enteric strains. CV (E1118 and E.TW09308) strains were more difficult to classify, with smaller margin of majority votes especially for E.TW09308, where the vote decision for environmental source is 46.46% and 53.54% for gastrointestinal. This observation is in line with the characteristics of CV strains that have been reported to lead a dual lifestyle, retaining the ability to survive in the external environments without a loss in the ability to colonize gastrointestinal tracts [34]. All the corresponding accession number and background information of these *Escherichia* sp. strain used in this study can be found in Table A1.

### 3.5. Function Analysis of Top Predictor Genes

The 164 top predictor genes were assigned to 18 COG categories based on functional annotation. Apart from those with function unknown (S), a large proportion of them was involved in five COG categories, namely inorganic ion transport and metabolism (P), energy production and conversion (C), amino acid transport and metabolism (E), carbohydrate transport and metabolism (G), and transcription (K) (Figure 4).

KEGG pathway analysis showed that the predictor genes mainly involved in five pathways: metabolic pathways, biosynthesis of secondary metabolites, two-component system, microbial metabolism in diverse environments and ABC transporters (Table 3).

Functions of 110 predictor genes from the top predictors were more conserved in the strains from external environment compared to only 45 genes that were more conserved in gastrointestinal strains (Mood’s median test, Benjamini–Hochberg critical value for false discovery rate = 0.25). Notably, around 50% of these 155 genes were present in all strains from both groups. One example of such predictor genes that showed subtle divergence between the two groups (i.e., with small bitscore difference) is the dcm gene that encodes the enzyme DNA-cytosine methyltransferase, which catalyze the highly conserved pathway of cytosine DNA methylation [38]. Genes that were enriched in the environmental strains included yihM (encodes a putative TIM barrel domain-containing protein YihM), yhiF (encodes a putative LuxR family regulatory protein), and genes encoding 1,2-propanediol utilization proteins, i.e., phosphotransacylase PduL and CoA-acylating propionaldehyde dehydrogenase PduP. Contrarily, genes that were enriched in the gastrointestinal strains included the ycjW and yidL genes (encoding putative DNA-binding transcriptional regulators), yidJ (putative sulfatase/phosphatase YidJ), and ynfG (anaerobic dimethyl sulfoxide (DMSO) reductase chain YnfG).

## 4. Discussion

In this study, we analyzed the genome contents of representative cryptic clade II strains of *Escherichia* to look for evidences of ecological differentiation within their population. Little is known of the evolution, habitat and lifestyle of theses bacteria, which were first recovered at very low frequency from non-human hosts and then later repeatedly recovered from external environment [18,20,23,39]. Dissecting the population level diversity of clade II in the context of ecology is not straightforward, as it involved a complex interplay between genetic variation (generated through mutations and gene flow), natural selection and genetic drifts, not to mention the confounding effects of factors such as geographical limitations [40]. The clade II strains formed a monophyletic cluster with high bootstrap support [18], we observed that the clade II strains of host gastrointestinal origins were also more distantly related to those isolated from external environment (Figure A2). Intuitively, the observed sequence disparity likely stemmed from their habitat difference (gastrointestinal vs. external environment). However, the observed distinction could also be attributed to biogeographic effects [1,10]. Apart from being isolated from two distinct categories of environments, the available genomes for this group of cryptic *Escherichia* also represented strains from different geographical regions. The strains from inter-tidal sediment were from Hong Kong, whereas the host-associated strains were from Australia, the United States, and Gabon [18,20,23,39]. Moreover, as only very few strains were isolated from hosts, these might represent spillover events instead of being sources of clade II strains (low frequency of recovery could also be due to less sampling efforts/difficulties in obtaining samples from wild animals). It is also possible that the discovery of these strains from both environments reflected a lifestyle that is similar to cryptic clade V of *Escherichia*, which were able to colonize gastrointestinal tracts while retaining traits favoring their survival in external environments [34,36].

Thus, to better understand the genetic diversity of these cryptic clade II strains in relation to their ecology, we analyzed the genomes of selected strains for interpretable difference that would indicate if these strains were indeed ecologically different, in terms of their association with two very distinct habitats i.e., host gastrointestinal and external environment. Leveraging on the ability of the combination of DBS metrics and random forest classifier to detect functional convergence in bacterial lineages with independent parallel adaptation to similar habitats or ecological niches [8,13], we adopted a similar approach to confirm if the strains indeed showed genomic signatures of adaptation to distinct habitats that correspond to isolation source. The approach could detect a spectrum of genetic changes that imply functional divergence, ranging from subtle functional change to gene deletion [8,13]. Inclusion of subtle changes (reflected by small bitscore differences), which may be reflective of diversifying selection, especially enhanced the sensitivity of the method [8,13]. Around half of the genes that were informative of habitat association in this study were such genes that are present in all genomes but displayed small bitscore differences between the gastrointestinal and external environment groups. The first manifestation of divergence between gastrointestinal and environmental clade II strains came from DBS metrics, where the gastrointestinal strains exhibited a general pattern of greater gene degradation, a common phenomenon in bacterial lineages that were more host-adapted and led a less generalist lifestyles [41]. Building on this observation, we trained a random forest classifier to classify the clade II strains according to their expected habitats (of gastrointestinal origins or from external environment). As the ecological distinctness of the clade II strains were preliminary at this stage, we guided the model with inclusion of other *Escherichia* genomes that has exhibited traits of adaptation to one of the habitats. Such genomes included *S. dysenteriae* from the *Shigella* genus, which is essentially a lineage of *E. coli* that had evolved from multiple *E. coli* lineages to become highly specialized human pathogens through convergent evolution involving independent horizontal gene transfers and gene losses [35].

Inclusion of these genomes that are distantly related to clade II in the training of random forest classifier increased the likelihood of scoring predictor genes that are relevant to parallel adaptations to these habitats, while minimizing the odds of classification by predictors that were more informative of phylogeny [13]. The benefit of such approach become apparent when contrasted with model built solely on the clade II strains. Using a set of 4922 orthologous genes, we performed the same model building process as described earlier for the six clade II genomes, resulting in a model that distinguished the two groups based on 762 predictor genes (Figure A3 and Figure A4, Appendix A), which was a much larger set and more difficult to interpret compared to the 164 genes identified by the original trained model (Model 6 in result section). Accuracy of the trained model was confirmed by its correct prediction of the habitats of an independent set of strains with known habitat association. It was also through comparison with this additional data set that the possibility of clade II being similar to clade V (i.e., capable of dual lifestyle) were largely dismissed (Figure 3). 

The trained model returned a set of predictor genes, many of which were indicative of the ecological divergence of these strains. Here, a few distinct examples were discussed. Consistent with previous findings, genes involved in 1,2-propanediol utilization were more enriched in the environmental *Escherichia* genomes. The pduL gene (phosphotransacylase PduL) was also previously showed to be enriched in environmental genomes (*Escherichia* cryptic clades III, IV, and V) [3]. Nonetheless, we found that the *pduP* gene (CoA-acylating propionaldehyde dehydrogenase PduP) was present in all tested cryptic clade II strains and only absent in *S. dysenteriae* Sd197 and *E. coli* ECOR66 but were differentiated by lower bitscore among the gastrointestinal strains. Notably, the *dcm* gene that encodes the enzyme DNA-cytosine methyltransferase was present in all environmental and gastrointertinal genomes but with sufficient bitscore difference between the two groups. These enzymes contained a highly variable region, which functions to recognize diverse DNA sequences, distinguishing own DNA from unwanted foreign DNAs or phage attack on basis of specific methylation patterns [42,43]. The bitscore divergence between the two groups for this gene could be taken as an indication of their association with two very different habitats, with different risks to tackle. Genes that were enriched in the gastrointestinal strains included *ynfG* (anaerobic DMSO reductase chain YnfG) and *yidJ* (putative sulfatase/phosphatase YidJ) that were previously shown to be essential or beneficial for survival in host gastrointestinal environments. The ability to use DMSO as electron receptor in anaerobic respiration may provide additional advantages for survival and persistence in host intestines [44]. On the other hand, sulfatase genes have been vital for the survival, competitive fitness, and host colonization of commensal such as Bacteroidetes and pathogenic bacteria such as *Salmonella* [45,46]. Sulfatase genes greatly enhanced the degradation of highly sulfated colonic mucins by catalyzing desulfation, enabling foraging of colonic mucins for carbohydrates [45]. Further verification of the accuracy and usefulness of these and other predictor genes in defining adaptation to host and external environment would be needed. Nonetheless, our results provided a starting point to identify candidate functions or pathways that are likely to display differential patterns in functional analysis experiments to confirm distinctness of these ecologically differentiated groups.

Furthermore, our work demonstrated the applicability of the DBS and machine learning approach beyond its original application on the niche adaptation of Salmonella enterica and expected application on any other pathogenic bacteria [13]. We anticipated the broader utilization of this approach on ecological studies and broader fields beyond, for the universality of the fundamental principle of this approach, i.e., genetic variations that are indicative of functional changes, which apply to all organisms.

## 5. Conclusions

Altogether, our study manifested that the cryptic clade II of *Escherichia* constituted strains that diverged to occupy host and external environments. Despite limited genome availability, the analyzed genomes displayed signatures of functional divergence that are consistent with adaptation to these two distinct environments. We outlined a subset of genes that were indicative of the habitat they were likely adapted to, which could guide selection of pathways or functions for further ecological studies to confirm such ecological distinctness. A clearer and comprehensive picture of the diversity and population structure of the clade II *Escherichia* would emerge in the future when more strains are recovered from different habitats, especially from animal hosts. With a larger collection of strains from diverse sources, further refinement of the gene subset (and their functions) that define adaptation to diverse habitats could be done using approaches similar to the current study. Furthermore, the genetic background that underlies their ecological differentiation could be dissected at higher resolution, probing the subtle yet crucial functional divergences that are fundamental to adaptation to a variety of hosts and external environments.

## Figures and Tables

**Figure 1 microorganisms-08-01713-f001:**
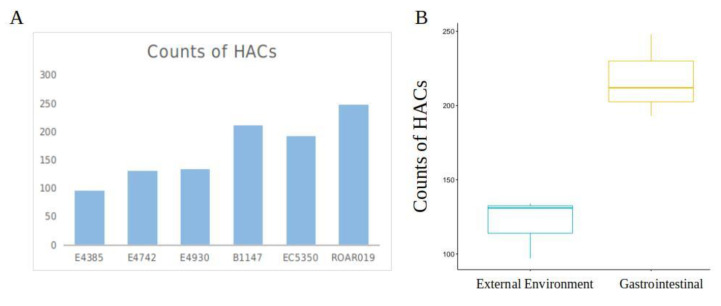
(**A**) Counts of HACs for each strain. First three strains (E4385, E4742 and E4930) were isolated from coastal marine sediment (external environmental), and the latter three (B1147, EC5350 and ROAR019) were of fecal origins (gastrointestinal). (**B**) Boxplot for one-way ANOVA analysis of the difference in number of HACs between the two groups of strains.

**Figure 2 microorganisms-08-01713-f002:**
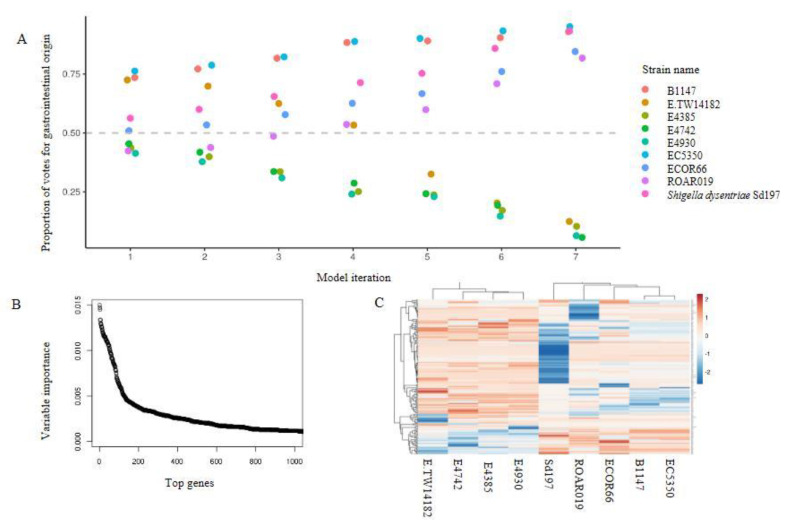
A subset of *Escherichia* spp. genes strongly indicates the existence of two adaptation phenotypes. (**A**) Casting of out-of-bag votes for isolation source of each strain by each model. The dashed grey line represents the voting threshold to classify a strain as of gastrointestinal origin. Model 1 utilized all predictor genes, and subsequent model iterations were built using sparsity pruning from predictor genes of preceding iteration. The sixth iteration achieved 100% accuracy for distinguishing the two groups, with majority votes of at least 70%. (**B**) Variable importance for the top 1000 genes that were used in initial training (model 1). Around 200 genes display high importance in distinguishing gastrointestinal vs environmental strains. (**C**) Heatmap of the top 164 predictor genes based on their bitscore value. Rows are centered and unit variance scaling is applied to rows with standard deviation as scaling factor. Imputation is used for missing value estimation. Rows and columns are clustered using correlation distance and average linkage (https://biit.cs.ut.ee/clustvis/). The color scale reflects the bitscore of respective strain for each orthologous gene. The more the negative value, the greater the deviation from reference protein in eggNOG database.

**Figure 3 microorganisms-08-01713-f003:**
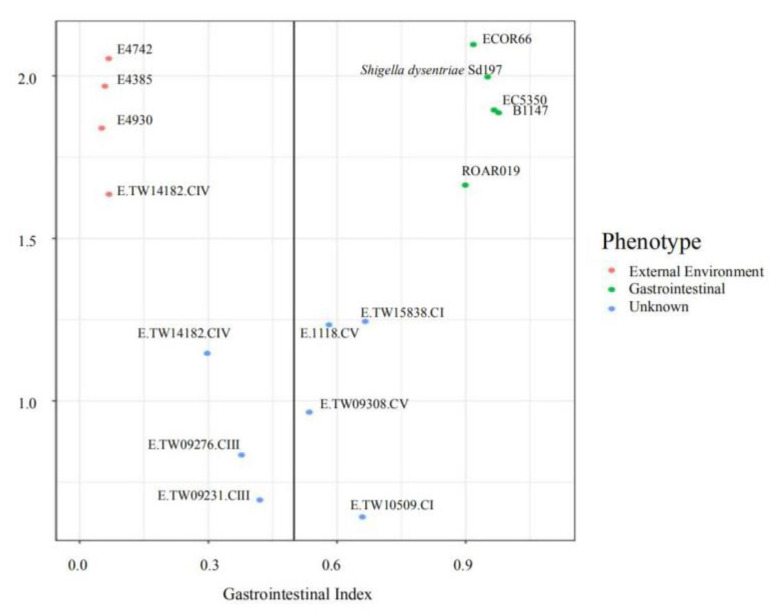
Isolation source prediction by the random forest model. The phenotype “External Environment” and “Gastrointestinal” represented strains from each source that were used in model training. The “Unknown” phenotype represents strains that were not included in the training dataset. The vertical line represents the voting threshold to separate the strains into of external environment or gastrointestinal origin. Data points to the left of the threshold were predicted to be from external environment, while those to the right were predicted to be of gastrointestinal origin.

**Figure 4 microorganisms-08-01713-f004:**
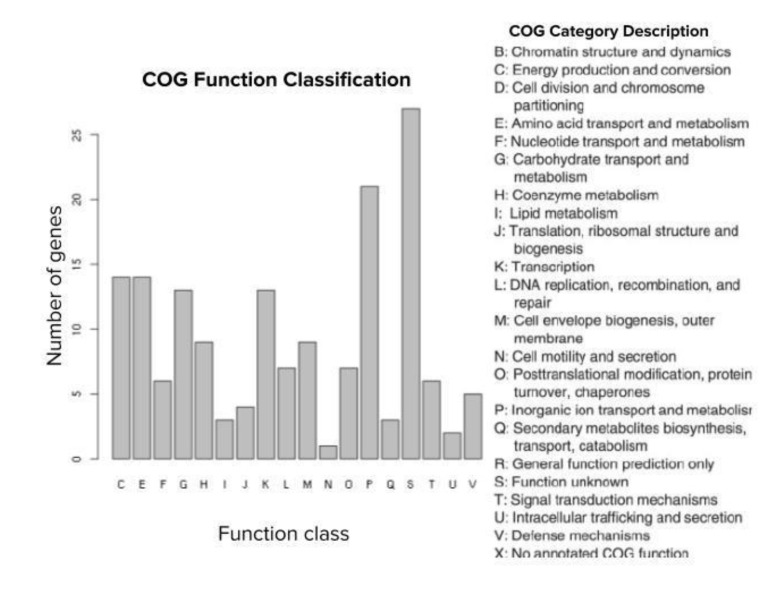
COG function classification of the top predictor genes. A total of 164 predictor genes showed homology to the COG database with the COG classification among 18 categories.

**Table 1 microorganisms-08-01713-t001:** *p* Values for the skewness of DBS values distribution from pairwise comparison between strains.

	ROAR019	EC5350	E4742	E4385	E4930
B1147	0.681	0.258	**0.002**	**0.05**	**0.014**
ROAR019		0.989	**0.001**	**0.042**	**0.014**
EC5350			**0.004**	**0**	**0.012**
E4742				0.25	0.644
E4385					0.256

Bold font indicated statistically significant (*p* < 0.05) skewness of DBS values distribution.

**Table 2 microorganisms-08-01713-t002:** Accuracy of permuted model.

	True Isolation Source			Assigned Isolation Source		
Strain Name	Source	Vote Proportion as G	Random1	Vote Proportion as G	Random2	Vote Proportion of G
E.TW14182	E	0.2025	E	0.3985	G	0.7962
E4385	E	0.1715	**E**	**0.5211**	**G**	0.3274
E4742	E	0.1932	G	0.7264	E	0.2668
E4930	E	0.1469	G	0.6551	E	0.2828
EC5350	G	0.9333	**E**	**0.7422**	**G**	**0.3272**
ROAR019	G	0.7089	G	0.6535	E	0.2702
Sd197	G	0.8590	G	0.6406	G	0.8781
B1147	G	0.9045	**G**	**0.3365**	G	0.6106
ECOR66	G	0.7608	E	0.3968	G	0.7832

E: external environment G: gastrointestinal; bold represented vote decision conflicting with the assigned isolation source.

**Table 3 microorganisms-08-01713-t003:** Main KEGG functional categories associated with top predictor genes. PERMANOVA test statistic (gastrointestinal vs. external environment strains) are shown for each category.

Number of Gene	Term	Degrees of Freedom	Sum of Squares	R2	F	Pr(>F)
30	Metablic pathways	1	0.0001	0.5950	10.2829	**0.025**
	Residual	7	0.0001	0.4050		
	Total	8	0.0002	1		
11	Biosynthsis of secondary metabolities	1	0.0129	0.2449	2.2712	0.098
	Residual	7	0.0399	0.7550		
	Total	8	0.0529	1		
10	Two-component system	1	0.0507	0.3641	4.0082	**0.038**
	Residual	7	0.0885	0.6358		
	Total	8	0.1392	1		
10	Microbial metabolism in diverse environments	1	0.0504	0.4299	5.2794	**0.026**
	Residual	7	0.0668	0.5701		
	Total	8	0.1172	1		
8	ABC transporters	1	0.0089	0.2794	2.7154	**0.005**
	Residual	7	0.0229	0.7205		
	Total	8	0.0317	1		
7	Biosynthesis of amino acids	1	0.0079	0.1743	1.4778	0.4
	Residual	7	0.0375	0.8257		
	Total	8	0.0454	1		
7	Biosynthesis of antibiotics	1	0.0329	0.3225	3.3329	**0.045**
	Residual	7	0.0691	0.6774		
	Total	8	0.1021	1		

Significant *p*-values (at a level of alpha = 0.05) are highlighted in bold.

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
