# Peer review of "Genetic Variation and Preliminary Indications of Divergent Niche Adaptation in Cryptic Clade II of Escherichia"

_microorganisms, 2020, doi:10.3390/microorganisms8111713_

Round 1
Reviewer 1 Report
Dear Editor,
I have carefully read the submission entitled Genetic variation and signatures of divergent niche adaptation in cryptic clade II of Escherichia by Zhi and colleagues. The authors analysed the genomes of selected strains of Escherichia coli on the basis of indications of ecological differentiation within their population. In this work they coupled the delta bitscore metrics to identify functional divergence of orthologous genes and the random forest classifier to differentiate the genomes according to habitats (external environment or gastrointestinal origin).
The presentation of the manuscript is adequate and the analysis of the data is appropriate. However, the main concern regards the dataset used by authors that is limited and thus does not strongly support conclusions. The same authors are concerned about this. However, these findings can be considered preliminary and this should be clear already in the title as well as in the abstract.
As minor comment, please correct the genus and species names must be written in Italic.
The manuscript is written with an acceptable level of English language.
Author Response
Reviewer’s comments:
I have carefully read the submission entitled Genetic variation and signatures of divergent niche adaptation in cryptic clade II of Escherichia by Zhi and colleagues. The authors analysed the genomes of selected strains of Escherichia coli on the basis of indications of ecological differentiation within their population. In this work they coupled the delta bitscore metrics to identify functional divergence of orthologous genes and the random forest classifier to differentiate the genomes according to habitats (external environment or gastrointestinal origin).
The presentation of the manuscript is adequate and the analysis of the data is appropriate. However, the main concern regards the dataset used by authors that is limited and thus does not strongly support conclusions. The same authors are concerned about this. However, these findings can be considered preliminary and this should be clear already in the title as well as in the abstract.
As minor comment, please correct the genus and species names must be written in Italic.
The manuscript is written with an acceptable level of English language.
Point 1: The presentation of the manuscript is adequate and the analysis of the data is appropriate. However, the main concern regards the dataset used by authors that is limited and thus does not strongly support conclusions. The same authors are concerned about this. However, these findings can be considered preliminary and this should be clear already in the title as well as in the abstract.
Response 1: Please provide your response for Point 1.
Thank you for your suggestion. We have made a few changes accordingly to indicate that the findings are preliminary. We have changed the title of the manuscript to “Genetic variation and preliminary indications of divergent niche adaptation in cryptic clade II of Escherichia”.
In the abstract, we used “preliminary indications” to replace “indications” (line 3). We also changed the last sentence (line 13-14) accordingly.
Original sentence:
“In summary, ecological differentiation was evident within the cryptic clade II of Escherichia, with strains displaying genomic signatures that are consistent with divergent adaptation to gastrointestinal and external environments.”
Revised sentence (line 13-14):
In summary, the cryptic clade II of Escherichia displayed genomic signatures that are consistent with divergent adaptation to gastrointestinal and external environments.
Point 2: As minor comment, please correct the genus and species names must be written in Italic.
Response 2:
We have corrected all the genus and species names to Italic format in the main text (title, lines 1, 7, 13, 16, 20, 22, 23, 24, 52, 60, 69, 75, 78-81, 133, 134, 159, 235, 261, 273, 279, 302-304, 323, 324, 326, 340, 347, 352, 358).
Similar corrections have been made for the reference section.
We have also italicized all the gene names (line 251, 253, 256-258, 323, 325, 328, 334, 335).
Reviewer 2 Report
The manuscript by Zhi and colleagues entitled “ Genetic variation and signatures of divergent niche adaptation in cryptic clade II of Escherichia” represents a starting point for the evaluation of gene-prediction model able to discriminate different ecotypes of bacteria.
- Major comments: the methodology used has a potential to be applied in the ecology field. However, the species considered in this study is limited and thus I suggest to put more emphasis on the methods used. In particular, the authors should add more details in the materials and methods section to allow the scientific community to apply this method. Moreover, authors might indicate if this informative gene-prediction model can be extended to different/wider field of research.
- Minor comments: The paper is well written although several grammar errors were found. In addition, scientific names of the organisms considered in this study have to be italicized. Please check in the text and in the reference section
Author Response
Reviewer’s comments:
The manuscript by Zhi and colleagues entitled “ Genetic variation and signatures of divergent niche adaptation in cryptic clade II of Escherichia” represents a starting point for the evaluation of gene-prediction model able to discriminate different ecotypes of bacteria.
Point 1: Major comments: the methodology used has a potential to be applied in the ecology field. However, the species considered in this study is limited and thus I suggest to put more emphasis on the methods used. In particular, the authors should add more details in the materials and methods section to allow the scientific community to apply this method. Moreover, authors might indicate if this informative gene-prediction model can be extended to different/wider field of research.
Response 1:
Thank you for your suggestion, we have revised section 2.5 and 2.6 for more clarity on the methods used. Moreover, we have included a link containing the pipelines that were used to calculate DBS and to construct the random forest classifier (line 113-167).
- 5. DBS Calculation and Definition of Hypothetically Attenuated Coding Sequences (HACs)
We retrieved the Pfam HMMs collection from ftp://ftp.ebi.ac.uk/pub/databases/Pfam/current_release, and use the curated Pfam-A.hmm database in this study. Annotated protein sequences (from section 2.4) were aligned to corresponding profile HMM in the Pfam database using hmmsearch from the HMMER3.0 package (http://hmmer.org) to produce bitscore values for each query sequence.
When comparing the proteomes of two strains, for two orthologous sequences (each from one strain) aligned to the same profile HMM, delta-bitscore (DBS) was calculated as follow [8]:
|
DBS = Xs1 – Xs2
|
(1) |
- where Xs1, and Xs2 each stand for bitscore of strain 1 and strain 2 respectively. DBS was calculated for all orthologues in such manner.
- Subsequently, for pairwise comparison of two proteomes, loss-of-function mutations were
- identified using empirical distributions of DBS values for all orthologues, with 2.5% of DBS on the
- least dispersed end as cutoff as described in Wheeler et al. [8]. Proteins with scores falling outside
- the cutoff are considered as HACs, i.e. hypothetically attenuated coding sequences. For concurrent
- comparison of multiple proteomes, as in Wheeler et al. [8], bitscores from all compared proteomes were
- first sorted. Then, the median score was used as benchmark, i.e. Xs1 in equation (1), to compare the
- bitscore of each gene to obtain DBS. Subsequently, HACs were identified from the DBS distribution using
- the same cutoff as pairwise comparison above (2.5% of DBS on the least dispersed tail). Wilcoxon signed
- rank test and one-way ANOVA were performed to test for significant difference in the degree of
- functional loss between two proteomes and between two groups of proteomes (gastrointestinal vs
- environmental), respectively. Analyses were done using the Minitab® 1 software (Minitab Inc,
- Pennsylvania, United States). Boxplot for ANOVA was created via the ggpubr package in R.
- 6. Random Forest Classifier Construction
- As the clade II strains were from two isolation sources with disparate environmental conditions
- (external environment vs gastrointestinal), we proceed to train a random forest classifier model to
- separate the strains based on their isolation sources, which presumably would return a set of genes
- that were indicative of adaptations to the different habitats. To minimize the inclusion of genes
- that were phylogenetic markers rather than informative of phenotype, we included another three
- more distantly related genomes, i.e. Shigella dysenteriae Sd197, coli ECOR66 (both gastrointestinal)
- and Escherichia clade IV TW14182 (environmental). Previous studies on these genomes showed
- evidences of adaptations to either external environment or gastrointestinal habitats for the genomes [3,
- 21,22,31–36]. Functional importance of sequence variations were scored using the DBS metric by
- comparing the protein coding genes of each strain to the profile HMMs of Gammaproteobacterial
- proteins (gproNOG.hmm) from the eggNOG database (http://eggnogdb.embl.de/). As we have
- performed with Pfam A in above pairwise comparisons, each protein sequence was searched against
- the gproNOG.hmm database using hmmsearch from HMMER3.0 package.
- In this study, orthologs were selected using filtering criteria similar to Wheeler et al. [13].
- We trained the random forest model on a set of 6,709 orthologous genes to differentiate strains
- of external environment and gastrointestinal origins, with the performance of model assessed by
- out-of-bag (OOB) accuracy. The random forest classifier was built and trained using the R packages
- “randomForest” and “caret”. Tree parameters were tested to evaluate the best combinations. Number
- of trees (ntree) were set at 10,000 to optimize mtry (number of genes randomly sampled at as candidates
- at each node) as OOB error rate stabilized at this ntree. We tested different values for mtry (1, n/10,
- n/5, n/3, n/2 and n, where n = the number of predictor genes) and decided on mtry=n/10 as it would
- reduce the chances of sampling correlated predictors and with lower OOB error (OOB error=0.2).
- Model performance was sequentially improved through several rounds of iterative feature selection [37].
- At the first iteration (model 1), all predictor genes were included in building the model, followed by sparsity
- pruning of predictors with variable importance (VI) ≤ The mod https://doi.org/10.1023/A:1010933404324 pruning of predictor genes with
- lowest 50% of VI) until perfect OOB accuracy was obtained. We performed permutation testing to test
- the null hypothesis of random association between the predictor genes and response variables
- (gastrointestinal or external environment). Model building pipeline was repeated on 1000 permutated
- datasets where the response variables were randomized. The p value, i.e. the frequency of models
- with the same accuracy as that of the original data was determined. The final model was used to predict
- the isolation source of another set of seven cryptic Escherichia clades genomes with known isolation
- sources to assess the accuracy of the trained classifier model. CIII (E. TW14182 and E. TW11588)
- and CIV (E.TW09276 and E. TW09231) strains showed signatures of adaptation to aquatic
- environments, CI (E. TW15838 and E. TW10509) strains were enteric, and CV (E1118 and E.
- TW09308) strains have been reported to retain the ability to survive in the external environments
- without loss the ability to in persist in gastrointestinal tracts [3, 21,22,31,33,34,36]. The top predictor
- genes of the best model were assigned to functional categories based on COG and KEGG database
- annotation matching via online web server KOBAS3.0 (http://kobas.cbi.pku.edu.cn/anno_iden.php).
- Summary of the pipelines used to obtain DBS and to construct the random forest classifier can be found at https://gist.github.com/szypanther/0cd83513e07aa9e1f9929f5df9214864.
At the last paragraph of the discussion (line 342-349), we indicated the possibility of applying the approach used in this and previous studies to wider field of research.
- Further verification of the accuracy and usefulness of these and other predictor genes in defining
- adaptation to host and external environment would be needed. Nonetheless our results provided a
- starting point to identify candidate functions or pathways that are likely to display differential patterns in
- functional analysis experiments to confirm distinctness of these ecologically differentiated groups. Furthermore,
- our work demonstrated the applicability of the DBS and machine learning approach beyond its original
- application on the niche adaptation of Salmonella enterica and expected application on any other pathogenic
- bacteria [13]. broader fields beyond, for the universality of the fundamental principle of this approach, i.e.
- genetic variations that are indicative of functional changes, which apply to all organisms.
Point 2: Minor comments: The paper is well written although several grammar errors were found. In addition, scientific names of the organisms considered in this study have to be italicized. Please check in the text and in the reference section
Response 2:
We have corrected all the genus and species names to Italic format in the main text (title, lines 1, 7, 13, 16, 20, 22, 23, 24, 52, 60, 69, 75, 78-81, 133, 134, 159, 235, 261, 273, 279, 302-304, 323, 324, 326, 340, 347, 352, 358).
Similar corrections have been made for the reference section.
We have also italicized all the gene names (line 251, 253, 256-258, 323, 325, 328, 334, 335).
Manuscript was re-examined for grammatical errors and indicated in the manuscript with “track changes” function enabled.
Reviewer 3 Report
In my opinion, the manuscript entitled “ Genetic variation and signatures of divergent niche adaptation in cryptic clade II of Escherichia” submitted by Lau and co-workers provide a valuable tool to discriminate E. coli belonging to cryptic clade II to unravel if clade II is comprised of strains that were ecologically divergent, using a novel approach.
The authors performed a combined approach based on delta bit-score values and random forest classifier to identify genes that are informative of Escherichia coli niche adaptation to hosts and external environments.
The results obtained in this study contributes get information on the specific function of each ecotypes with potential application in the field of ecology. Thus, I’m inclined towards the positive side.
I have only some suggestion that should be considered by the authors:
-please provide a more detailed explanation of how DBS has been calculated comparing the reference and the variant strain. Note that I’m referring to the beginning of paragraph 2.5 until the definition of each term of the equation used for the calculation.
I don’t know if it is correct but, the DBS has been calculated following these steps:
Annotated proteins (derived from the search using BUSCO??) were aligned to the corresponding HMM in the Pfam database using hmmsearch to produce bit-score values? This is not clear.
A possible idea will be to provide (also as supplementary material) a summarizing scheme of the pipeline used to arrive to the definition of predictor genes to facilitate the reproducibility of the method used, that have potential to be extended also outside the field of microorganisms.
-please, uniform the use of the terms “bit-score” and “bitscore” in the main text (line 108).
-please, use the italic to mention all the scientific names of the organisms considered in this study, also in the references section.
-the name of genes, these have to be corrected using the italic (lines 255, 257, 260, 261, 262, 327, 329, 332, 338, 339).
-The English is quite good but I recommend a thorough rereading of the text.
Author Response
In my opinion, the manuscript entitled “ Genetic variation and signatures of divergent niche adaptation in cryptic clade II of Escherichia” submitted by Lau and co-workers provide a valuable tool to discriminate E. coli belonging to cryptic clade II to unravel if clade II is comprised of strains that were ecologically divergent, using a novel approach.
The authors performed a combined approach based on delta bit-score values and random forest classifier to identify genes that are informative of Escherichia coli niche adaptation to hosts and external environments.
The results obtained in this study contributes get information on the specific function of each ecotypes with potential application in the field of ecology. Thus, I’m inclined towards the positive side.
I have only some suggestion that should be considered by the authors:
Point 1: -please provide a more detailed explanation of how DBS has been calculated comparing the reference and the variant strain. Note that I’m referring to the beginning of paragraph 2.5 until the definition of each term of the equation used for the calculation.
I don’t know if it is correct but, the DBS has been calculated following these steps:
Annotated proteins (derived from the search using BUSCO??) were aligned to the corresponding HMM in the Pfam database using hmmsearch to produce bit-score values? This is not clear.
Response 1:
Thank you for your suggestion, we have revised section 2.5 for more clarity on the methods used (line 113-126).
- 5. DBS Calculation and Definition of Hypothetically Attenuated Coding Sequences (HACs)
We retrieved the Pfam HMMs collection from ftp://ftp.ebi.ac.uk/pub/databases/Pfam/current_release, and use the curated Pfam-A.hmm database in this study. Annotated protein sequences (from section 2.4) were aligned to corresponding profile HMM in the Pfam database using hmmsearch from the HMMER3.0 package (http://hmmer.org) to produce bitscore values for each query sequence.
When comparing the proteomes of two strains, for two orthologous sequences (each from one strain) aligned to the same profile HMM, delta-bitscore (DBS) was calculated as follow [8]:
|
DBS = Xs1 – Xs2
|
(1) |
- where Xs1, and Xs2 each stand for bitscore of strain 1 and strain 2 respectively. DBS was calculated for all orthologues in such manner.
- Subsequently, for pairwise comparison of two proteomes, loss-of-function mutations were
- identified using empirical distributions of DBS values for all orthologues, with 2.5% of DBS on the
- least dispersed end as cutoff as described in Wheeler et al. [8]. Proteins with scores falling outside
- the cutoff are considered as HACs, i.e. hypothetically attenuated coding sequences. For concurrent
- comparison of multiple proteomes, as in Wheeler et al. [8], bitscores from all compared proteomes were
- first sorted. Then, the median score was used as benchmark, i.e. Xs1 in equation (1), to compare the
- bitscore of each gene to obtain DBS. Subsequently, HACs were identified from the DBS distribution using
- the same cutoff as pairwise comparison above (2.5% of DBS on the least dispersed tail). Wilcoxon signed
- rank test and one-way ANOVA were performed to test for significant difference in the degree of
- functional loss between two proteomes and between two groups of proteomes (gastrointestinal vs
- environmental), respectively. Analyses were done using the Minitab® 1 software (Minitab Inc,
- Pennsylvania, United States). Boxplot for ANOVA was created via the ggpubr package in R.
Point 2: A possible idea will be to provide (also as supplementary material) a summarizing scheme of the pipeline used to arrive to the definition of predictor genes to facilitate the reproducibility of the method used, that have potential to be extended also outside the field of microorganisms.
Response 2:
Thank you for your suggestion. We have included a link containing the pipelines that were used to calculate DBS and to construct the random forest classifier (line 167).
- Summary of the pipelines used to obtain DBS and to construct the random forest classifier can be found at https://gist.github.com/szypanther/0cd83513e07aa9e1f9929f5df9214864.
Point 3: -please, uniform the use of the terms “bit-score” and “bitscore” in the main text (line 108).
Response 3:
We have uniformed the use of “bitscore” in the whole paper.
Point 4: -please, use the italic to mention all the scientific names of the organisms considered in this study, also in the references section.
Response 4:
We have corrected all the genus and species names to Italic format in the main text (title, lines 1, 7, 13, 16, 20, 22, 23, 24, 52, 60, 69, 75, 78-81, 133, 134, 159, 235, 261, 273, 279, 302-304, 323, 324, 326, 340, 347, 352, 358).
Similar corrections have been made for the reference section, i.e. lines 372-474.
Point 5: -the name of genes, these have to be corrected using the italic (lines 255, 257, 260, 261, 262, 327, 329, 332, 338, 339).
Response 5:
We have also italicized all the gene names in the revised manuscript (lines 251, 253, 256-258, 323, 325, 328, 334, 335).
Point 6:-The English is quite good but I recommend a thorough rereading of the text.
Response 6:
Manuscript was re-examined for grammatical errors and typos, and were indicated in the manuscript with “track changes” function enabled.